# Light-inducible antimiR-92a as a therapeutic strategy to promote skin repair in healing-impaired diabetic mice

Tina Lucas[1,2,*], Florian Schäfer[3,*], Patricia Müller[3], Sabine A. Eming[4,5,6], Alexander Heckel[3,**] & Stefanie Dimmeler[1,2,**]

MicroRNAs (miRs) are small non-coding RNAs that post-transcriptionally control gene expression. Inhibition of miRs by antisense RNAs (antimiRs) might be a therapeutic option for many diseases, but systemic inhibition can have adverse effects. Here we show that light-activatable antimiRs efficiently and locally restricted target miR activity *in vivo*. We use an antimiR-92a and establish a therapeutic benefit in diabetic wound healing. AntimiR-92a is modified with photolabile protecting groups, so called 'cages'. Irradiation activates intradermally injected caged antimiR-92a without substantially affecting miR-92a expression in other organs. Light activation of caged antimiR-92a improves healing in diabetic mice to a similar extent as conventional antimiRs and derepresses the miR-92a targets Itga5 and Sirt1, thereby regulating wound cell proliferation and angiogenesis. These data show that light can be used to locally activate therapeutically active antimiRs *in vivo*.

[1] Institute of Cardiovascular Regeneration, Centre for Molecular Medicine, Goethe University Frankfurt, Theodor-Stern-Kai 7, Frankfurt 60590, Germany. [2] German Center for Cardiovascular Research (DZHK), RheinMain Oudenarder Str. 16, Berlin 13347, Germany. [3] Institute for Organic Chemistry and Chemical Biology, Buchmann Institute for Molecular Life Sciences, Goethe University Frankfurt, Max-von-Laue-Straße 15, Frankfurt 60438, Germany. [4] Department of Dermatology, University of Cologne, Kerpenerstr. 62, Cologne 50937, Germany. [5] Cologne Excellence Cluster on Cellular Stress Responses in Aging-Associated Diseases (CECAD), University of Cologne, Joseph-Stelzmann-Str. 26, Cologne 50931, Germany. [6] Center for Molecular Medicine Cologne (CMMC), University of Cologne, Robert-Koch-Str. 21, Cologne 50931, Germany. * These authors contributed equally to this work. ** These authors jointly supervised this work. Correspondence and requests for materials should be addressed to S.D. (email: dimmeler@em.uni-frankfurt.de).

MicroRNAs (miRs) are small non-coding RNAs that post-transcriptionally regulate gene expression by binding to targeted mRNAs and thereby inducing degradation or blocking its translation. MiRs have important functions in different pathophysiological processes and diseases. Therefore, targeting miRs by application of specific miR inhibitors might have great therapeutic potential. MiRs can be inhibited by different types of antisense RNAs (antimiRs) including phosphothioate-modified antisense RNAs that are linked to cholesterol for enhanced uptake of the oligonucleotide or locked nucleotide acids DNA mixmers[1]. Such antisense oligonucleotides are relatively easily taken up by detoxifying organs such as the liver or kidney, but uptake in other organs such as the muscle or brain tends to be limited[2,3]. Local delivery or activation may be necessary to augment the biological functions of antimiRs in the target tissue and reduce systemic toxicity. Local activation might also avoid unwanted side effects of antimiRs, since miRs have diverse functions in different tissues. Several targeting strategies have been experimentally used including the linking of miRs or antimiRs to aptamers[4,5], nanoparticle- or microparticle-mediated delivery[6,7] and cell type-specific delivery by viral vectors[8] as well as attempts of local delivery by mechanical tools, for example, catheters[9]. In addition, we and others have developed photoactivatable antimiRs by attaching photolabile protecting groups (cages) to the nucleobases that temporarily inhibit duplex formation with the target miR, thereby allowing an excellent on/off behaviour upon irradiation[10–13]. However, the therapeutic *in vivo* use of light-activatable antimiRs has been unclear. Therefore, we tested whether light-activatable antimiRs directed against miR-92a can be used to locally augment impaired wound healing in diabetic mice. Inhibition of miR-92a was previously shown to improve angiogenesis and recovery after ischaemia[9,14,15]; however, its regulation and function during wound healing, a process that is dependent on the angiogenic response, is unknown. Here we show that light-activatable antimiR-92a efficiently downregulate miR-92a expression leading to target gene derepression in the murine skin, thereby improving diabetic wound healing by stimulating cell proliferation and angiogenesis.

## Results

**Light-inducible antimiR activation in the murine skin.** In order to transfer the concept of activation with light *in vitro* to non-transparent tissue, we established an *ex vivo* skin explant culture model to define suitable wavelengths, light dose and concentration of applied caged antimiRs. First, we used isolated skin tissue of adult male mice and measured light intensity behind a skin barrier by testing light-emitting diodes (LEDs) of different wavelengths and currents. These findings were compared to light intensities without the skin barrier. Although longer wavelengths show, as expected, a better penetration, 385 nm light efficiently enters and penetrates the murine skin (Supplementary Fig. 1). Since wavelengths of this range can effectively remove photolabile-protecting groups[10] but are not expected to induce direct photo damage as compared to shorter wavelengths of the ultraviolet spectrum, light of 385 nm was used for the subsequent experiments. Caged antimiR-92a, developed in earlier studies[10], was injected intradermally into murine skin explants (1 μg diluted in 50 μl PBS). After irradiation for 10 min (385 nm, 300 mA) and subsequent cultivation for 48 h, levels of miR-92a were analysed by quantitative real-time PCR. In comparison to the samples treated with non-irradiated caged antimiR-92a, miR-92a levels are significantly downregulated upon light induction (Fig. 1a). The efficiency of miR-92a inhibition by light-induced caged antimiRs is similar as

compared to conventional constitutively active antimiRs, which are used as positive control (Fig. 1a). A caged control antimiR with no target-specific sequences, which is used as negative control, does not inhibit miR-92a expression neither in the presence nor absence of light (Fig. 1a).

For *in vivo* studies, we synthesized antimiRs with a slightly changed architecture, in accordance with first antagomiR studies by Krutzfeldt *et al.*[16] (Supplementary Fig. 2). AntimiRs were composed of 2′-OMe-modified nucleotides and contained in total six phosphorothioates at the 3′- and 5′-end to allow stability against endonucleases and exonucleases. We attached a cholesterol moiety on the 3′-end of the antimiRs for better *in vivo* delivery. To temporarily abolish antimiR activity, we attached six 1-(2-nitrophenyl)ethyl photolabile-protecting groups to the $N^6$ of A and $N^4$ of C nucleotides (Supplementary Fig. 2), respectively. Additionally, we synthesized a non-caged antimiR-92a as a positive control and a caged antimiR as a negative control that does not affect miR-92a levels (Supplementary Fig. 2). The purity of the compounds was confirmed by gel electrophoresis and analytical high-performance liquid chromatography and we demonstrated the efficient removal of cage groups after light activation *in vitro* (Supplementary Fig. 3).

These antimiRs were then injected using the same concentrations, injection strategy and light activation as described above. The *in vivo* activation of caged antimiR-92a leads to a significant reduction of miR-92a levels in the murine skin with an efficiency that is comparable to constitutively active antimiR-92a, while the controls have no effects (Fig. 1b). Additionally, by measuring miR-92a level in the kidney and liver, we show that locally light-activated antimiR-92a does not significantly affect the kidney or liver miR-92a levels (Fig. 1c,d). These data demonstrate that caged antimiRs can be successfully activated by light *in vivo*. Local activation results in reduced systemic inhibitory effect of the antimiRs and as such may reduce adverse effects in other organs.

**miR-92a inhibition improves wound healing in diabetic mice.** Cutaneous wound healing is a highly orchestrated physiological process involving the interplay of keratinocytes, fibroblasts, endothelial cells, extracellular matrix remodeling and the immune system[17]. Inflammation, proliferation and remodeling, the three different healing phases, that overlap in time and space, are regulated not only by several cytokines and growth factors but also by various miRs[18]. The formation of new blood vessels by proliferation and migration of endothelial cells (a process termed angiogenesis) promotes the wound-healing response[19,20]. AntimiR-92a augments angiogenesis in muscle tissues after ischaemia[9,14]; however, its role in the complex process of wound healing is unknown. Therefore, we measured miR-92a level in acute wounds from Bl/6 wild-type mice and healing-impaired diabetic wounds from db/db mice (time points were chosen in accordance with the estimated peak of angiogenesis) as well as in acute human wound tissue and from non-healing human venous leg ulcers. These analysis revealed that miR-92a is strongly upregulated in chronic wounds in comparison to acute wounds, indicating that miR-92a may represent a therapeutic target to rescue impaired chronic wound healing (Fig. 2a).

Next, we tested the therapeutic applicability of light-activated antimiR-92a. To this end, we used diabetic mice as a relevant model of impaired wound healing. Non-healing, chronic wounds are a particularly common problem in diabetes. Db/db mice are deficient for leptin receptor activity by carrying a homozygous point mutation in the gene for this receptor and are a well-established model to test therapeutic improvements for chronic wound healing[21]. To study skin repair with or without functional

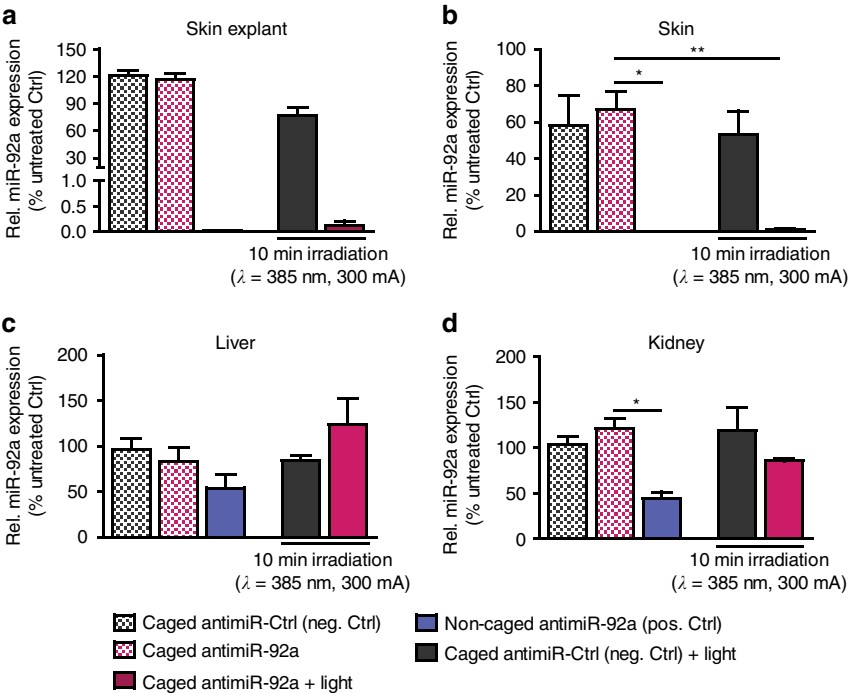

**Figure 1 | Efficient miR-92a downregulation in the skin upon treatment with caged antimiR-92a.** (**a**) miR-92a expression level quantified by real-time PCR in skin explants after 48 h of intradermal injection with different antimiRs as indicated ± light activation *ex vivo*. $n = 2$ explants per treatment, isolated from two different mice. qPCR analysis for miR-92a levels in the skin (**b**), liver (**c**) and kidney (**d**) upon i.d. injection of different antimiRs as indicated ± light activation *in vivo*. RNA was isolated after 48 h of injection. $n = 4$ different animals per group. Data are expressed as means ± s.e.m. Significance of difference was analysed using analysis of variance one-way test analysis with Newman–Keuls Multiple Comparison Test. **$P < 0.01$, *$P < 0.05$.

miR-92a in diabetic mice, 13-week-old mice were wounded by using 6 mm punch biopsies and wounds were treated with caged antimiRs (2 µg diluted in 25 µl PBS) at days 0, 4 and 7. Knockdown efficiency was quantified via qPCR 6 and 11 days post injury. Light-activated caged antimiR-92a therapy leads to a marked downregulation of miR-92a levels whereas no effect is seen in control groups (Supplementary Fig. 4).

In order to analyse the distribution of caged antimiRs, we injected Cy3-labelled caged antimiR-92a and isolated the wound tissue 6 days post injury. (Supplementary Fig. 5). Positive cells are detected in the hyperproliferative epithelium and within the dermal compartment. Moreover, double staining shows that Cy3-labelled antimiRs are taken up by F4/80-positive macrophages (Supplementary Fig. 5c).

The wound-healing kinetics were observed over a period of 11 days and the wound size was quantified macroscopically at defined days post injury. While in the early healing phase, no differences are observed among the different groups macroscopically, both the conventional antimiR-92a and the caged antimiR-92a with light activation exhibit a smaller wound size 11 days post injury (Fig. 2b). In contrast to control antimiR-treated wounds, light-induced antimiR-92a or constitutively active antimiR-92a-treated wounds lose scabs 11 days post-wounding (Fig. 2c), indicative of accelerated re-epithelialization. These findings were further confirmed by microscopic analysis of haematoxylin and eosin-stained paraffin sections isolated at days 6 and 11 post injury. Although almost all wounds are closed at day 11 postwounding, miR-92a inhibitor-treated wounds reveal faster wound closure kinetics, estimated by a shorter distance between epithelial tips measured at day 6 postwounding (Fig. 2d). Furthermore, wounds after antimiR-92a treatment show a more dense and cell-rich granulation tissue, equipped with a thick and hyperproliferative epithelium at day 11 postwounding (Fig. 2e,g). In contrast, control wounds have a more fragile appearance and

are only covered by a thin epithelial layer, often detached from the underlying granulation tissue (Fig. 2g). Finally, wound contraction, estimated by measuring the distance between the edges of the panniculus carnosus, are significantly stronger upon miR-92a inhibition at day 11 post injury (Fig. 2f), indicating that caged antimiR-92a harbour great therapeutic potential to improve wound healing in diabetic mice.

To rule out toxic side effects of repetitive caged antimiR treatments and the released 2-nitrosoacetophenon side product upon light activation, plasma levels for the liver transaminases aspartate aminotransferase and alanine aminotransferase as well as for the alkaline phosphatase were measured as biomarkers for liver injury. Furthermore, creatinine and urea plasma levels were determined to control for kidney function. For all used biomarkers, no significant upregulation is detected in comparison to untreated db/db mice (Supplementary Fig. 6), indicating that caged antimiRs and the released photolysis side product have no obvious toxic short-term effects on organs, such as the liver and kidney.

**AntimiR-92a enhances cell proliferation and angiogenesis.** In order to analyse the underlying mechanism for improved healing upon miR-92a inhibition, we first characterized the wound-healing response by measuring angiogenesis, wound cell proliferation, myofibroblast differentiation and the inflammatory response post injury. Indeed, a significant higher amount of CD31[+] blood vessels within the granulation tissue is observed in mice that are treated with light-activated antimiR-92a and constitutively active antimiR-92a as compared to all the control groups at days 6 and 11 post injury (Fig. 3a,b). Furthermore, staining for the proliferation marker Ki67 reveals improved wound cell proliferation. Quantification of positive stained cells unveils that upon inhibition of miR-92a proliferation is improved

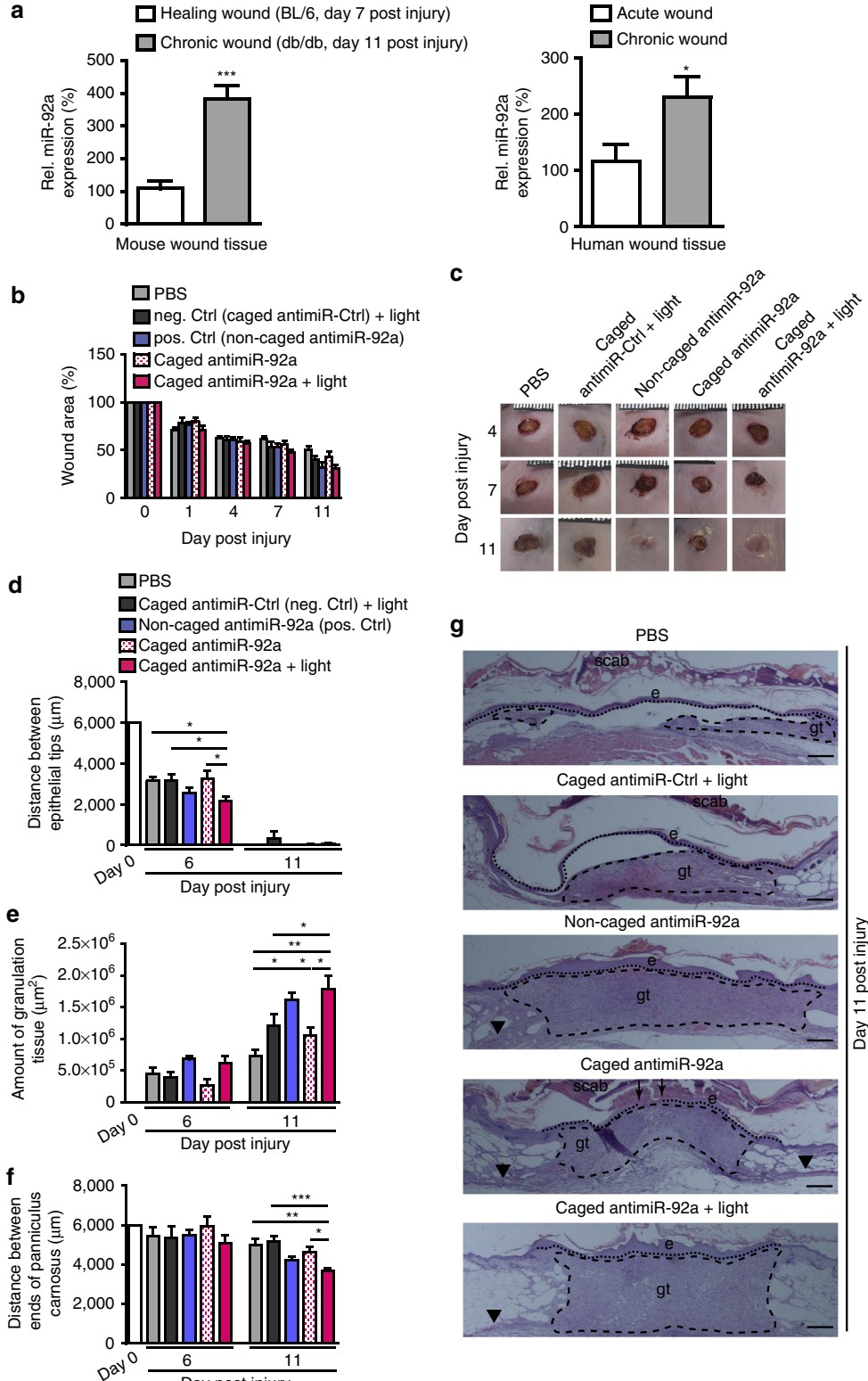

**Figure 2 | Caged antimiR-92a treatment improves wound healing in db/db mice.** (**a**) miR-92a expression level quantified by real-time PCR in murine and human wound tissue as indicated. (**b**) Macroscopic quantification of the wound size over time postwounding. Different groups as indicated. (**c**) Representative photographs of the quantified wound area at different time points post injury. Morphometric quantification of (**d**) the length of reepithelialization, as a measure for the wound size; (**e**) the amount of granulation tissue, as a measure for wound quality; and (**f**) the distance between the ends of panniculus carnosus, as a measure for wound contraction. (**g**) Representative haematoxylin and eosin stainings of wound tissue 11 days post injury. Different groups as indicated. $n = 4$–14 wounds per group on 3–7 different mice. Dotted line marks epithelial dermal border and dashed line highlights the granulation tissue. Arrows point to epithelial tips and arrow heads to the end of panniculus carnosus. e, hyperproliferative epithelium; gt, granulation tissue. Scale bars, 300 µm. Data are expressed as means ± s.e.m. Significance of difference was analysed using analysis of variance one-way test analysis with Newman–Keuls Multiple Comparison Test. ***$P < 0.001$, **$P < 0.01$, *$P < 0.05$.

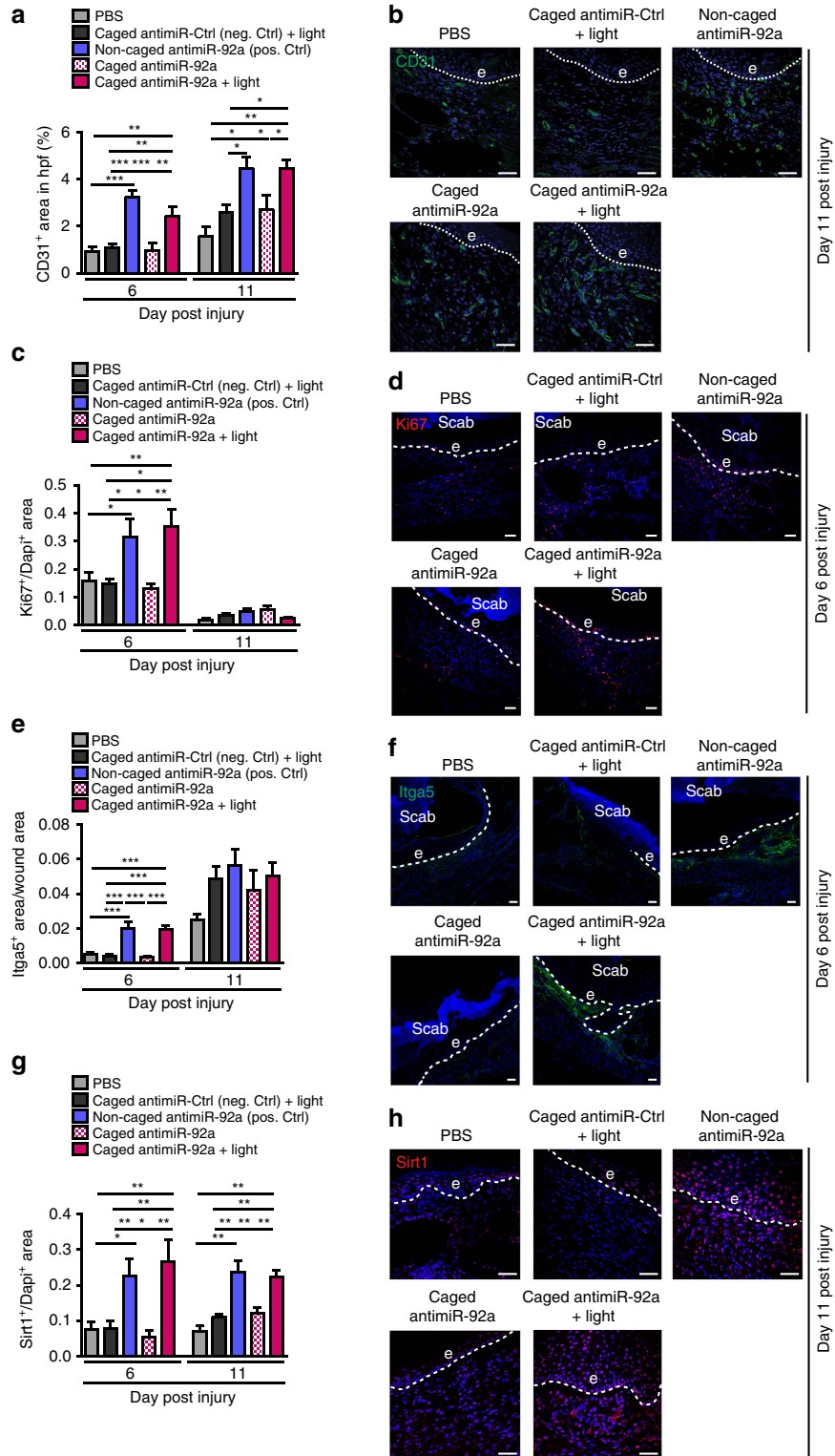

**Figure 3 | Inducible miR-92a inhibition increases angiogenesis and proliferation.** (**a**) Immunohistochemical quantification of the CD31-positive-stained area within the granulation tissue and (**b**) representative images of wound sections 11 days post injury stained for CD31. Different groups as indicated. $n = 4$–14 wounds per group on 3–7 different mice. Hpf, high power field. (**c**) Morphometric quantification of the Ki67$^+$ area normalized to the Dapi$^+$ area within the granulation tissue 6 and 11 days post injury. (**d**) Ki67 and Hoechst double staining of day 6 wounds. $n = 3$–10 wounds per group on 2–5 different mice. (**e**) Morphometric quantification of the Itga5$^+$ area normalized to the wound area at day 6 and 11 post injury. (**f**) Itga5 and Dapi double staining of day 6 wounds. (**g**) Morphometric quantification of the Sirt1$^+$ area normalized to the Dapi$^+$ area within the granulation tissue 6 and 11 days post injury. (**h**) Sirt1 and Hoechst double staining of day 11 wounds. Scale bars, 50 μm; e, hyperproliferative epithelium. Data are expressed as means ± s.e.m. Significance of difference was analysed using analysis of variance one-way test analysis with Newman–Keuls Multiple Comparison Test. ***$P < 0.001$, **$P < 0.01$, *$P < 0.05$.

within the epithelial and the dermal compartment at day 6 post injury (Fig. 3c,d). Besides angiogenesis and re-epithelialization, myofibroblast differentiation and the inflammatory response are two other essential events for successful healing. However, for both, we excluded an influence of miR-92a inhibition estimated by staining for the myofibroblast marker α-smooth muscle actin (αSMA; Supplementary Fig. 7a,b) and the pan inflammatory cell marker CD45 (Supplementary Fig. 7c,d).

Validated pro-angiogenic miR-92a targets are the integrin subunit α5 (Itga5) that mediates cell–matrix interactions and cell migration[22,23] via binding to fibronectin and the class III histone deacetylase sirtuin (Sirt)-1 (ref. 14). The upregulation of both was quantified by immunohistochemical stainings against Itga5 and Sirt1 of wound sections. While for Itga5 only during the early healing response an upregulation is detected (Fig. 3e,f), Sirt1 expression is significantly induced upon effective miR-92a inhibition at days 6 and 11 post injury, generally highlighting the functional inhibition of miR-92a upon caged antimiR-92a treatment following light activation (Fig. 3g,h). Besides the angiogenesis-related functions of Itga5 (ref. 14) and Sirt1 (refs 24,25) that might explain the healing-supportive effects of miR-92a inhibition, both proteins are also shown to control re-epithelialization.

Itga5 is expressed on endothelial and epithelial cells during the wound-healing response. An increase of Itga5 during the early phase of the wound-healing response is contributing to keratinocyte migration over the fibronectin-rich matrix. In line, we observe an upregulation of Itga5 during the early wound-healing phase in miR-92a-inhibited wounds probably mediating an improved keratinocyte migration and finally forcing the observed improved wound-closure kinetics at day 6 post injury (Fig. 3e,f). In addition, Itga5 abundance in dermal wound tissue is reflecting the angiogenic response since Itga5 is expressed on endothelial cells. In accordance with those findings, we find an upregulation of the Itga5 signal in dermal wound tissue in all groups at day 11 post injury with a slight stronger signal in antimiR-92a-treated samples (Fig. 3e).

The second miR-92a target Sirt1 was shown to promote corneal epithelialization[26] and improved keratinocyte proliferation during skin repair[27]. In line, a strong upregulation of Sirt1 upon miR-92a inhibition in the epithelium covering the wound area is observed (Fig. 3h), probably leading to the thick and hyperproliferative epithelium in miR-92a-inhibited wounds and thereby promoting an accelerated healing response (Fig. 2). Together, we find that antimiR-92a-treated wounds reveal faster healing kinetics based on a stronger angiogenic response and increased wound cell proliferation mediated by the derepression of the miR-92a targets Itga5 and Sirt1.

In conclusion, here we show that caged antimiRs are effectively activated by light in non-transparent living tissue. To the best of our knowledge, the concept of ultraviolet uncaging of oligonucleotides in the skin of living mice has only been shown once before for photoregulation of aptamer activity and biodistribution[28] and to uncage green fluorescent protein-coding plasmids[29]. Furthermore, there has been no approach to spatially confined therapeutic use of antimiRs by light activation in vivo. Inhibiting miR-92a by light-activatable antimiRs may harbour great therapeutic potential to treat chronic skin repair in healing impaired diabetic patients avoiding unwanted adverse effects in other tissues that might be expected by systemic delivery. However, given the low potential of cytotoxicity of systemic miR-92a treatment, our study can be rather seen as 'poof-of-concept' experiment paving the way to testing cytotoxic antimiRs for treating malignant diseases or antimiRs that could be valuable for inducting regeneration, which often have the risk for promoting tumor growth. In addition, future experiments will show if the concept of light activation can be transferred to the

application of well-perfused inlying organs—eventually necessitating visible or near-infrared light deprotection—and thereby expand the therapeutic benefit for different antimiR therapies in the future, including acute myocardial infarction or cancer.

## Methods

**Chemical synthesis.** All reactions were performed under argon atmosphere using dry solvents. 2′-OMe nucleoside precursors were purchased from Carbosynth. The synthesis of the phosphoramidite precursors were performed analogously to ref. 10 as follows (see Supplementary Figs 8 and 9 for an overview):

**Synthesis of 5′-O-(4,4′-dimethoxytrityl)-3′-O-(tertbutyldimethylsilyl)-2′-O-methyluridine (2).** 5′-O-(4,4′-dimethoxytrityl)-2′-O-methyluridine (500 mg, 0.89 mmol, 1 equiv.) and imidazole (152 mg, 2.23 mmol, 2.5 equiv.) were dissolved in dimethylformamide (DMF, 8 ml) and tert-butyldimethylsilyl chloride (250 mg, 1.66 mmol, 1.86 equiv.) was added, and the reaction was stirred at room temperature for 45 h. Upon addition of ethanol (20 ml), the reaction was stopped, the solvent was removed and $CH_2Cl_2$ was added. The organic layer was washed consecutively with 5% aqueous citric acid solution and saturated aqueous $NaHCO_3$ solution and dried over $MgSO_4$, filtered and concentrated. The residue was purified by column chromatography (cyclohexane/ethyl acetate 1:1 v/v) to afford product 2 (569 mg, 0.84 mmol, 95%) as a colourless foam. $^1$H-NMR (400 MHz, DMSO-$d_6$): $\delta = 11.39$ (d, $J = 2.1$ Hz, 1H, NH), 7.86 (d, $J = 8.1$ Hz, 1H, $H_{ar}$), 7.38–7.30 (m, 4H, $H_{ar}$), 7.27–7.22 (m, 5H, $H_{ar}$), 6.91–6.88 (m, 4H, $H_{ar}$), 5.79 (d, $J = 2.6$ Hz, 1H, 1′-H), 5.29 (dd, $J = 8.1$ Hz, $J = 2.1$ Hz, 1H, $H_{ar}$), 4.33–4.30 (m, 1H, 3′-H), 3.93–3.89 (m, 1H, 4′-H), 3.86–3.84 (m, 1H, 2′-H), 3.73 (s, 6H, $OCH_3$), 3.42–3.38 (m, 4H, 5′-H, $OCH_3$), 3.21–3.17 (m, 1H, 5′-H), 0.76 (s, 9H, $SiC(CH_3)_3$), 0.03 (s, 3H, $SiCH_3$), $-0.05$ (s, 3H, $SiCH_3$) p.p.m. $^{13}$C-NMR (62.9 MHz, DMSO-$d_6$): $\delta = 163.0$, 158.2, 150.2, 144.4, 140.4, 135.2, 135.0, 129.8, 129.8, 127.8, 127.8, 126.8, 113.2, 101.4, 87.7, 86.0, 82.2, 82.2, 69.5, 61.8, 57.7, 55.0, 25.5, 17.6, $-4.7$, $-5.3$ p.p.m. Matrix-assisted laser desorption/ionization–high-resolution mass spectrometry (MALDI-HRMS): $m/z$ calculated for $C_{37}H_{46}N_2O_8Na$ $[M + Na]^+$ 697.29156, found 697.29297 ($\Delta m = 0.0014$, error 2 p.p.m.).

**Synthesis of 5′-O-(4,4′-dimethoxytrityl)-3′-O-(tertbutyldimethylsilyl)-4-O-(2,4,6-triisopropylbenzenesulfonyl)-2′-O-methyluridine (3).** Compound 5 (296 mg, 0.44 mmol, 1 equiv.) and 4-dimethylaminopyridine (DMAP; 5 mg, 0.04 mmol, 0.1 equiv.) were dissolved in $CH_2Cl_2$ (4 ml). Then diisopropylethylamine (380 µl, 2.19 mmol, 5 equiv.) was added and the reaction mixture was cooled to 0 °C. Triisopropylbenzenesulfonic acid chloride (240 mg, 0.792 mmol, 1.8 equiv.) was added and the mixture was stirred for 15 min at 0 °C and then for 90 min at room temperature. By addition of saturated $NaHCO_3$ solution, the reaction was stopped. The aqueous layer was extracted with $CH_2Cl_2$ and the organic phase was dried over $MgSO_4$, filtered and concentrated. The crude product was purified by column chromatography (cyclohexane/ethyl acetate 6:1 → 5:1 → 4:1 v/v) to afford product 3 (414 mg, quantitative) as a colourless foam. $^1$H-NMR (400 MHz, CD$_3$CN): $\delta = 8.58$ (d, $J = 7.3$ Hz, 1H, $H_{ar}$), 7.40–7.38 (m, 2H, $H_{ar}$), 7.35 (s, 2H, $H_{ar}$), 7.32–7.25 (m, 7H, $H_{ar}$), 6.87–6.85 (m, 4H, $H_{ar}$), 5.76–5.75 (m, 1H, 1′-H), 5.64 (d, $J = 7.3$ Hz, 1H, $H_{ar}$), 4.36–4.33 (m, 1H, 3′-H), 4.20 (sep, $J = 6.8$ Hz, 2H, CH), 4.06–4.02 (m, 1H, 4′-H), 3.77 (s, 6H, $OCH_3$), 3.73–3.72 (m, 1H, 2′-H), 3.57 (dd, $J = 11.3$ Hz, $J = 2$ Hz, 1H, 5′-H), 3.53 (s, 3H, $OCH_3$), 3.33 (dd, $J = 11.3$ Hz, $J = 2.8$ Hz, 1H, 5′-H), 2.97 (sep, $J = 6.9$ Hz, 1H, CH), 1.29–1.24 (m, 18H, 6 × $CH_3$), 0.77 (s, 9H, $SiC(CH_3)_3$), 0.03 (s, 3H, $SiCH_3$), $-0.06$ (s, 3H, $SiCH_3$) p.p.m. $^{13}$C-NMR (62.9 MHz, DMSO-$d_6$): $\delta = 168.0$, 160.0, 159.9, 156.2, 154.5, 152.0, 148.5, 145.5, 136.3, 136.2, 131.9, 131.3, 131.3, 129.2, 129.0, 128.2, 125.4, 114.2, 95.3, 90.8, 87.7, 84.2, 83.5, 69.8, 61.7, 58.9, 56.0, 35.0, 30.5, 26.0, 24.7, 23.7, 18.6, $-4.2$, $-4.9$ p.p.m. MALDI-HRMS: $m/z$ calculated for $C_{52}H_{68}N_2O_{10}SiNa$ $[M + Na]^+$ 963.42561, found 963.42242 ($\Delta m = 0.0032$, error 3.3 p.p.m.).

**Synthesis of 5′-O-(4,4′-dimethoxytrityl)-3′-O-(tertbutyldimethylsilyl)-4-N-(1-(2-nitrophenyl)ethyl)-2′-O-methylcytidine (4a).** Compound 3 (5.68 g, 6 mmol) was dissolved in DMF (80 ml). 1-(2-Nitrophenyl)ethylamine (1.74 g, 9.65 mmol, 1.6 equiv.) was added and the solution was stirred overnight at room temperature. An incomplete conversion of the starting material was detectable. Therefore, the reaction mixture was heated to 75 °C for 5 h and then stirred for 5 additional days at room temperature. The solvent was removed under reduced pressure and $CH_2Cl_2$ was added. The organic layer was washed consecutively with 5% aqueous citric acid solution and then saturated aqueous $NaHCO_3$ solution and dried over $MgSO_4$, filtered and concentrated to afford the crude product, which was purified by column chromatography ($CH_2Cl_2$/ethyl acetate 2:1 → 1:1 v/v). The product 4a (4.03 g, 4.89 mmol, 82%) was obtained as a light yellow foam. $^1$H-NMR (400 MHz, DMSO-$d_6$): $\delta = 8.43$–8.40 (m, 1H, NH), 7.96–7.89 (m, 2H, $H_{ar}$), 7.77–7.59 (m, 2H, $H_{ar}$), 7.53–7.46 (m, 1H, $H_{ar}$), 7.38–7.30 (m, 4H, $H_{ar}$), 7.28–7.21 (m, 5H, $H_{ar}$), 6.99–6.81 (m, 4H, $H_{ar}$), 5.77–5.71 (m, 1H, 1′-H), 5.61–5.58 (m, 1H, $H_{ar}$), 5.48–5.54 (m, 1H, CH), 4.31–4.25 (m, 1H, 3′-H), 3.91–3.88 (m, 1H, 4′-H), 3.74 (s, 6H, $OCH_3$), 3.62–3.59 (m, 1H, 2′-H), 3.46–3.43 (m, 1H, 5′-H), 3.41–3.37

(m, 3H, OCH$_3$), 3.14–3.10 (m, 1H, 5′-H), 1.51–1.47 (m, 3H, CH3), 0.72–0.71 (m, 9H, SiC(CH$_3$)$_3$), − 0.01 to − 0.12 (m, 6H, SiCH$_3$) p.p.m. $^{13}$C-NMR (62.9 MHz, DMSO-d$_6$): δ = 162.4, 158.2, 154.3, 154.2, 148.2, 148.1, 144.4, 140.2, 140.0, 139.8, 139.7, 135.2, 135.1, 135.0, 133.7, 129.8, 129.7, 128.0, 127.8, 127.7, 127.4, 126.7, 124.1, 113.2, 113.2, 94.1, 88.4, 85.9, 83.1, 82.8, 81.4, 69.3, 69.1, 57.6, 55.0, 45.1, 25.5, 21.7, 17.6, − 4.8, − 5.4 p.p.m. MALDI-HRMS: *m/z* calculated for C$_{45}$H$_{54}$N$_4$O$_9$SiNa [M + Na]$^+$ 845.35523, found 845.35158 (Δm = 0.00365, error 4.3 p.p.m.).

**Synthesis of 5′-O-(4,4′-dimethoxytrityl)-4-N-(1-(2-nitrophenyl)ethyl)-2′-O-methylcytidine (4b).** Compound **4a** (3.83 g, 4.65 mmol, 1 equiv.) was dissolved in tetrahydrofuran (THF; 80 ml) and a solution of tetra-n-butylammonium fluoride (TBAF; 1 M in tetrahydrofuran 13.9 ml, 13.9 mmol, 3 equiv.) was added dropwise. The reaction mixture was stirred for 90 min at room temperature. The solvent was removed and the crude product was purified by column chromatography (CH$_2$Cl$_2$/methanol 99:1 → 98:2 → 97:3 → 96:4 v/v) to afford product **4b** (3.13 g, 4.42 mmol, 95%) as a brownish foam. $^1$H-NMR (400 MHz, DMSO-d$_6$): δ = 8.42 (d, J = 7.1 Hz, 1H, NH), 7.95–7.91 (m, 1H, H$_{ar}$), 7.81–7.60 (m, 3H, H$_{ar}$), 7.53–7.46 (m, 1H, H$_{ar}$), 7.40–7.31 (m, 4H, H$_{ar}$), 7.27–7.25 (m, 5H, H$_{ar}$), 6.92–6.90 (m, 4H, H$_{ar}$), 5.78–5.73 (m, 1H, 1′-H), 5.62–5.58 (m, 1H, H$_{ar}$), 5.55–5.48 (m, 1H, CH), 5.11–5.07 (m, 1H, 3′-OH), 4.21–4.12 (m, 1H, 3′-H), 3.92–3.91 (m, 1H, 4′-H), 3.75 (s, 6H, OCH$_3$), 3.61–3.56 (m, 1H, 2′-H), 3.43–3.39 (m, 3H, OCH$_3$), 3.26–3.25 (m, 2H, 5′-H), 1.51–1.48 (m, 3H, CH$_3$) p.p.m. $^{13}$C-NMR (62.9 MHz, DMSO-d$_6$): δ = 162.4, 158.1, 154.3, 154.3, 148.2, 148.1, 144.7, 140.0, 139.8, 139.7, 135.4, 135.4, 135.2, 135.2, 133.7, 133.6, 129.7, 128.0, 128.0, 127.9, 127.7, 127.5, 127.4, 126.8, 124.1, 124.1, 113.2, 94.2, 94.1, 87.7, 87.5, 85.8, 83.4, 83.1, 81.7, 81.6, 68.3, 68.2, 62.2, 62.1, 57.7, 57.7, 55.0, 45.1, 45.0, 21.9, 21.7 p.p.m. MALDI-HRMS: *m/z* calculated for C$_{39}$H$_{40}$N$_4$O$_9$K [M + K]$^+$ 747.24269, found 747.24227 (Δm = 0.00042, error 0.6 p.p.m.).

**Synthesis of 3′-O-(2-cyanoethoxy-N,N-diisopropylamino)phosphine-5′-O-(4,4′-dimethoxytrityl)-4-N-(1-(2-nitrophenyl)ethyl)-2′-O-methylcytidine (5).** Compound **4b** (350 mg, 0.49 mmol, 1 equiv.) was dissolved in CH$_2$Cl$_2$ (10 ml). Diisopropylethylamine (430 µl, 2.47 mmol, 5 equiv.) was added and the solution stirred for 5 min at room temperature. 2-Cyanoethoxy-N,N-diisopropylamino-chlorophosphine (130 µl, 0.59 mmol, 2 equiv.) was added and the reaction mixture was stirred for 2 h at room temperature, and then CH$_2$Cl$_2$ was added. The organic layer was washed with saturated aqueous NaHCO$_3$ solution, dried over MgSO$_4$, filtered and concentrated to afford the crude product, which was purified by column chromatography (cyclohexane/acetone 3:1 → 2:1 → 1:1 v/v) to afford product **5** (202 mg, 0.22 mmol, 45%). $^1$H-NMR (400 MHz, DMSO-d$_6$): δ = 8.43–8.42 (m, 1H, NH), 7.95–7.79 (m, 2H, H$_{ar}$), 7.76–7.68 (m, 2H, H$_{ar}$), 7.65–7.59 (m, 1H, H$_{ar}$), 7.53–7.46 (m, 1H, H$_{ar}$), 7.40–7.31 (m, 4H, H$_{ar}$), 7.29–7.21 (m, 5H, H$_{ar}$), 6.92–6.87 (m, 4H, H$_{ar}$), 5.83–5.74 (m, 1H, 1′-H), 5.61–5.44 (m, 1H, H$_{ar}$), 5.51–5.46 (m, 1H, CH), 4.44–4.28 (m, 1H, 3′-H), 4.07–4.03 (m, 1H, 4′-H), 3.79–3.68 (m, 8H, OCH$_3$, 2′-H, 1 × OCH$_2$CH$_2$CN), 3.60–3.47 (m, 3H, 2 × CH, 1 × OCH$_2$CH$_2$CN), 3.42–3.36 (m, 4H, OCH$_3$, 1 × 5′-H), 3.27–3.20 (m, 1H, 1 × 5′-H), 2.77–2.74 (m, 1H, 1 × OCH$_2$CH$_2$CN), 2.60–2.55 (m, 1H, OCH$_2$CH$_2$CN), 1.50–1.49 (m, 3H, CH39, 1.14–0.92 (m, 12H, 4 × CH$_3$) p.p.m. $^{31}$P-NMR (121.4 MHz, DMSO-d$_6$): δ = 149.2, 149.0, 149.0 p.p.m.

**Synthesis of 3′-O-acetyl-5′-O-(4,4′-dimethoxytrityl)-2′-O-methylinosine (7).** 5′-O-(4,4′-Dimethoxytrityl)-2′-O-methylinosine (14.2 g, 24.3 mmol, 1 equiv.) and N,N-dimethylaminopyridine (314 mg, 2.43 mmol, 0.1 equiv.) were dissolved in pyridine (75 ml). Acetic acid anhydride (7.6 ml, 80.2 mmol, 3.3 equiv.) was added. The mixture was then stirred for 3 h at room temperature. By the addition of methanol (7 ml), the reaction was stopped. The solvent was removed and the residue was dissolved in CH$_2$Cl$_2$. The organic layer was washed with a solution of 5% aqueous citric acid and then with a saturated aqueous solution of NaHCO$_3$. The organic layer was dried over MgSO$_4$ filtered and concentrated to afford **1** in quantitative yield. $^1$H-NMR (250 MHz, acetone-d$_6$): δ = 11.19 (bs, 1H, NH), 8.12 (s, 1H, H2), 8.00 (s, 1H, H8), 7.50–7.46 (m, 2H, H$_{ar}$), 7.36–7.22 (m, 7H, H$_{ar}$), 6.88–6.85 (m, 4H, H$_{ar}$), 6.08 (d, J = 6.3 Hz, 1H, 1′-H), 5.56 (dd, J = 5.2 Hz, J = 3.4 Hz, 1H, 3′-H), 4.87–4.83 (m, 1H, 4′-H), 4.36–4.31 (m, 1H, 4′-H), 3.72 (s, 6H, 2 × OCH$_3$), 3.46 (d, J = 4.5 Hz, 2H, 2 × 5′-H), 3.36 (s, 3H, OCH$_3$), 2.11 (s, 3H, CH$_3$) p.p.m. $^{13}$C-NMR (62.9 MHz, DMSO-d$_6$): δ = 169.5, 158.1, 156.4, 148.1, 145.9, 144.6, 139.0, 135.3, 135.3, 129.7, 127.8, 127.6, 126.7, 124.8, 113.1, 85.7, 85.5, 81.5, 79.8, 70.6, 63.3, 58.2, 55.0, 20.6 p.p.m. MALDI-HRMS: *m/z* calculated for C$_{34}$H$_{35}$N$_4$O$_8$ [M + H]$^+$ 627.24498 found 627.24467 (Δm = 0.00031, error 0.5 p.p.m.).

**Synthesis of 3′-O-acetyl-5′-O-(4,4′-dimethoxytrityl)-6-O-(2,4,6-triisopropylbenzenesulfonyl)-2′-O-methylinosine (8).** Compound **1** (2.1 g, 3.35 mmol, 1 equiv.) and N,N-dimethylaminopyridine (314 mg, 0.35 mmol, 0.1 equiv.) were dissolved in CH$_2$Cl$_2$ (30 ml). The reaction mixture was cooled to 0 °C followed by the consecutive addition of diisopropylethylamine (2.9 ml, 16.8 mmol, 5 equiv.) and triisopropylbenzenesulfonic acid chloride (1.83 g, 6.04 mmol, 1.8 equiv.). After 10 min, the reaction mixture was allowed to warm to room temperature and was stirred for another 90 min. The mixture was then diluted with CH$_2$Cl$_2$ and washed with saturated aqueous NaHCO$_3$ solution. The organic layer was dried over

MgSO$_4$, filtered and concentrated. The oily residue was purified by column chromatography (cyclohexane/ethylacetate 6:1 → 4:1 → 2:1 v/v) to afford the product **2** (1.36 g, 1.52 mmol, 45%) as colourless foam and the N-sulfonylated side product (1.28 g, 43%). $^1$H-NMR (400 MHz, DMSO-d$_6$): δ = 8.74–8.73 (m, 1H, NH), 8.36 (bs, 1H, H2), 7.96–7.92 (m, 1H, H8), 7.89–7.85 (m, 2H, H$_{ar}$), 7.67–7.63 (m, 1H, H$_{ar}$), 7.44–7.40 (m, 1H, H$_{ar}$), 7.38–7.33 (m, 2H, H$_{ar}$), 7.27–7.15 (m, 7H, H$_{ar}$), 6.86–6.80 (m, 4H, H$_{ar}$), 6.02–5.96 (m, 1H, 1′-H), 5.80–5.74 (m, 1H, CH), 5.46–5.44 (m, 1H, 3′-H), 4.92–4.85 (m, 1H, 2′-H), 4.22–4.20 (m, 1H, 4′-H), 3.73–3.71 (m, 6H, OCH$_3$), 3.37–2.35 (m, 2H, 2 × 5′-H), 3.21 (bs, 3H, OCH$_3$), 2.09 (s, 3H, OCH$_3$), 1.65 (d, J = 6.9 Hz, 3H, CH$_3$) p.p.m. $^{13}$C-NMR (62.9 MHz, DMSO-d$_6$): δ = 158.1, 158.1, 158.1, 152.3, 152.2, 148.8, 144.7, 144.6, 140.1, 135.5, 135.3, 135.2, 133.4, 129.7, 129.7, 129.6, 127.8, 127.7, 127.6, 127.4, 126.7, 123.6, 113.1, 85.7, 85.7, 81.3, 79.5, 70.8, 70.7, 63.3, 58.1, 55.0, 20.7, 20.6 p.p.m.

**Synthesis of 3′-O-acetyl-5′-O-(4,4′-dimethoxytrityl)-6-N-(1-(2-nitrophenyl)-ethyl)-2′-O-methyladenosine (9a).** Compound **2** (1.32 g, 1.47 mmol, 1 equiv.) was dissolved in DMF (20 ml). 1-(2-Nitrophenyl(ethylamine) (426 mg, 2.36 mmol, 1.6 equiv.) was added dropwise to the reaction mixture. The reaction mixture was stirred for 1 day at room temperature and then for 1 day at 90 °C. 4-Dimethyla-minopyridine (18 mg, 0.15 mmol, 0.1 equiv.) was added and the reaction mixture was stirred for 1 day at room temperature and then for 5 h at 90 °C. N,N-Dime-thylaminopyridine (0.5 equiv.) was added because of the incomplete conversion of the starting material. The reaction mixture was stirred for 5 h at 90 °C and then for 60 h at room temperature. Upon completion, the solvent was removed, CH$_2$Cl$_2$ was added and the reaction mixture was washed consecutively with a solution of 5% aqueous citric acid and then a saturated aqueous solution of NaHCO$_3$. The organic layer was dried over MgSO$_4$, filtered and concentrated to afford the crude product, which was purified by column chromatography (cyclohexane/ethyl acetate 1:2 v/v) to afford the product **9a** (823 mg, 1.06 mmol, 72%) as a colourless foam. $^1$H-NMR (400 MHz, DMSO-d$_6$): δ = 8.74–8.73 (m, 1H, NH), 8.36 (bs, 1H, H2), 7.96–7.92 (m, 1H, H8), 7.89–7.85 (m, 2H, H$_{ar}$), 7.67–7.63 (m, 1H, H$_{ar}$), 7.44–7.40 (m, 1H, H$_{ar}$), 7.38–7.33 (m, 2H, H$_{ar}$), 7.27–7.15 (m, 7H, H$_{ar}$), 6.86–6.80 (m, 4H, H$_{ar}$), 6.02–5.96 (m, 1H, 1′-H), 5.80–5.74 (m, 1H, CH), 5.46–5.44 (m, 1H, 3′-H), 4.92–4.85 (m, 1H, 2′-H), 4.22–4.20 (m, 1H, 4′-H), 3.73–3.71 (m, 6H, OCH$_3$), 3.37–2.35 (m, 2H, 2 × 5′-H), 3.21 (bs, 3H, OCH$_3$), 2.09 (s, 3H, OCH$_3$), 1.65 (d, J = 6.9 Hz, 3H, CH$_3$) p.p.m. $^{13}$C-NMR (62.9 MHz, DMSO-d$_6$): δ = 158.1, 158.1, 158.1, 152.3, 152.2, 148.8, 144.7, 144.6, 140.1, 135.5, 135.3, 135.2, 133.4, 129.7, 129.7, 129.6, 127.8, 127.7, 127.6, 127.4, 126.7, 123.6, 113.1, 85.7, 85.7, 81.3, 79.5, 70.8, 70.7, 63.3, 58.1, 55.0, 20.7, 20.6 p.p.m.

**Synthesis of 5′-O-(4,4′-dimethoxytrityl)-6-N-(1-(2-nitrophenyl)ethyl)-2′-O-methyladenosine (9b).** An MeNH$_2$ solution (8 M in ethanol, 30 ml, 240 mmol, 67.2 equiv.) was added to compound **9a** (2.77 g, 3.57 mmol, 1 equiv.) and stirred overnight at room temperature. Ethanol and methylamine were removed under reduced pressure and the crude product was purified by column chromatography (cyclohexane/ethyl acetate 2:1 → 1:1 → 1:2 → 1:4 v/v) to afford the product **9b** (2.41 g, 3.29 mmol, 92%) as an yellow foam. $^1$H-NMR (400 MHz, DMSO-d$_6$): δ = 8.73–8.67 (m, 1H, NH), 8.35–8.31 (m, 1H, H2), 8.10–7.99 (m, 1H, H8), 7.89–7.86 (m, 2H, H$_{ar}$), 7.69–7.63 (m, 1H, H$_{ar}$), 7.46–7.34 (m, 3H, H$_{ar}$), 7.25–7.21 (m, 7H, H$_{ar}$), 6.87–6.80 (m, 4H, H$_{ar}$), 6.05–6.00 (m, 1H, 1′-H), 5.82–5.74 (m, 1H, CH), 5.27 (bs, 1H, 3′-OH), 4.46–4.41 (m, 2H, 3′-H, 2′-H), 3.74–3.73 (m, 7H, 4′-H, OCH$_3$), 3.34 (bs, 3H, OCH$_3$), 3.26–3.20 (m, 2H, 2 × 5′-H), 1.66 (d, J = 6.9 Hz, 3H, CH$_3$) p.p.m. $^{13}$C-NMR (75.5 MHz, DMSO-d$_6$): δ = 158.0, 158.0, 154.3, 153.2, 152.3, 152.2, 152.0, 150.0, 148.8, 144.8, 144.8, 144.7, 140.4, 139.9, 139.6, 135.5, 135.5, 135.3, 133.5, 129.8, 129.6, 127.7, 127.7, 127.6, 126.6, 123.6, 119.7, 113.1, 85.6, 85.5, 83.5, 81.8, 69.0, 63.5, 57.7, 57.7, 55.0, 44.7, 30.7, 21.4 p.p.m. MALDI-HRMS: *m/z* calculated for C$_{40}$H$_{41}$N$_6$O$_8$ [M + H]$^+$ 733.29804, found 733.29694 (Δm = 0.0011, error 1.5 p.p.m.).

**Synthesis of 3′-O-(2-cyanoethoxy-N,N-diisopropylamino)phosphine-5′-O-(4,4′-dimethoxytrityl)-6-N-(1-(2-nitrophenyl)ethyl)-2′-O-methyladenosine (10).** Diisopropylethylamine (145 µl, 0.83 mmol, 5 equiv.) was added to a solution of compound **3b** (122 mg, 0.167 mmol, 1 equiv.) in CH$_2$Cl$_2$ (4 ml), and the reaction mixture was stirred for 5 min at room temperature. Then 2-cyanoethoxy-N,N-diisopropylaminochlorophosphine (75 µl, 0.33 mmol, 2 equiv.) was added, and the mixture was stirred for another 90 min at room temperature. CH$_2$Cl$_2$ was added and the mixture was washed with saturated aqueous NaHCO$_3$ solution and dried over MgSO$_4$, filtered and concentrated to afford the crude product, which was purified by column chromatography (cyclohexane/acetone 3:1 → 2:1 v/v) to afford the product **10** (112 mg, 0.12 mmol, 72%) as an yellow foam. $^1$H-NMR (300 MHz, DMSO-d$_6$): δ = 8.69–8.68 (m, 1H, NH), 8.37–8.33 (m, 1H, H2), 7.98–7.92 (m, 1H, H8), 7.87–7.84 (m, 2H, H$_{ar}$), 7.67–7.62 (m, 1H, H$_{ar}$), 7.45–7.37 (m, 1H, H$_{ar}$), 7.35–7.29 (m, 2H, H$_{ar}$), 7.23–7.17 (m, 7H, H$_{ar}$), 6.84–6.79 (m, 4H, H$_{ar}$), 6.03–6.02 (m, 1H, 1′-H), 5.78–5.72 (m, 1H, CH), 4.69–4.60 (m, 2H, 3′-H, 2′-H), 4.19–4.08 (m, 1H, 4′-H), 3.83–3.77 (m, 1H, OCH$_2$CH$_2$CN), 3.72–3.70 (m, 6H, OCH$_3$), 3.64–3.48 (m, 3H, OCH$_2$CH$_2$CN, CH), 2.78–2.75 (m, 1H, OCH$_2$CH$_2$CN), 2.62–2.58 (m, 1H, OCH$_2$CH$_2$CN), 3.32–3.19 (m, 5H, OCH$_3$, 2 × 5′-H), 1.64 (d, J = 6.8 Hz, 3H, CH$_3$), 1.22–1.00 (m, 12H, CH$_3$) p.p.m. $^{31}$P-NMR (121.4 MHz, DMSO-d$_6$): δ = 149.4, 149.2 p.p.m.

Solid-phase synthesis of non-cholesterol-modified antimiRs was performed on an ABI 392 DNA/RNA synthesizer analogously to ref. 10 as follows: 2′-OMe RNA SynBaseTM 1000 Å CPG columns were used as solid support. Non-modified 2′-OMe RNA amidites were purchased from LinkTech. Synthesis was performed in DMTr-ON mode using modified synthesis protocols with 6 min coupling time for standard 2′-OMe amidites and 15 min coupling time for caged amidites. 5-Benzylthio-1H-tetrazole was used as an activator for caged oligonucleotides. For sulfurization, Beaucage's reagent was used. A modified synthesis cycle was used for thioate linkages with the capping step after sulfurization. After deprotection with conc. $NH_4OH$ (overnight at room temperature), the solvent was evaporated and the crude product was purified by RP-HPLC using the following columns: (a) CS Chromatography Nucleosil 100–5 C 18 ($4.6 \times 250$ mm), (b) Agilent Eclipse XDB-C18 ($4.6 \times 250$ mm), and (c) Nucleosil 5 C 18 ($20 \times 250$ mm). A dual buffer system with 0.1 M triethylammonium acetate pH 7 and acetonitrile was used with linear gradients from 5% to 56% acetonitrile depending on the column used and the hydrophobicity of the respective antimiR. The 5′-DMTr group was cleaved with 80% aqueous acetic acid (125 μl, 20 min at room temperature). After evaporation of the acetic acid, the residue was purified via RPHPLC using the same gradients as before. The purity and identity was confirmed by HPLC-ESI-MS (Bruker micrOTOF-Q): caged antimiR-92a: 5′-$C_sA_s$GGCCGGG$A$CAAGUGC$A_sA_sU_s$A-3′ (expected mass: 8062.5 Da, obtained mass: 8062.8 Da); caged antimiR-Ctrl: 5′-$A_sA_s$GGC$A$AGCUGACCCU$G$AA$_sG_sU_s$U-3′ (expected mass: 7896.4 Da, obtained mass: 7896.9 Da). The subscript 's' represents phosphorothioate linkages, bold script represents caged nucleotides.

Solid-phase synthesis of cholesterol-modified antimiRs was performed in 14 μmole scale on an automated DNA/RNA synthesizer (Äkta Oligopilot plus) in DMTr-OFF mode. 3′-Cholesteryl-TEG CPG and Cyanine 3 phosphoramidite were purchased from GlenResearch; 2′-OMe RNA amidites were supplied by LinkTech. The oligonucleotides were synthesized using a modified synthesis protocol with 6 min recycling time and a Cap-Ox-Cap or Cap-Thio-Cap protocol. Amidite concentrations were adjusted to 0.15 M; 5-ethylthiotetrazole (0.5 M) was used as an activator. Thiolation was achieved using a 0.05 M solution of 3-ethoxy-1,2,3-dithiazoline-5-one (EDITH). After deprotection with conc. $NH_4OH$ (overnight at room temperature), the caged antimiR was prepurified on an Äkta Purifier system with a 12 ml Source 15RPC column (flow: $4$ ml min$^{-1}$) using a two-step linear gradient with 0.1 M triethylammonium acetate (pH 7) in water (buffer A) and 0.1 M triethylammonium acetate in 90% acetonitrile (buffer B): Step 1: 0–20% buffer B in 1 column volume; Step 2: 20–90% buffer B in 8.4 column volumes.

Product-containing fractions were pooled and further purified on a semipreparative YoungLin SP930 HPLC system using a Multokrom 100–5 C18 column ($250 \times 20$ mm). A binary solvent system with 400 mM hexafluoroisopropanol (HFIP) and 16.3 mM triethylamine pH 7.9 (buffer C) and methanol (solvent D) with a flow rate of 8 ml min$^{-1}$ with the following two-step linear gradient was used: Step 1: 0–20% solvent D in 9 min; Step 2: 20–90% solvent D in 82.7 min

Non-caged antimiR-92a and caged antimiR-Ctrl were purified only by RP-HPLC on the YoungLin system.

After the cleavage from the solid support, the Cy3-labelled caged antimiR was prepurified on a MultoKrom 100 C18 column (250 mm × 4.6 mm) using a binary solvent system (buffer C and solvent D) and a two-step gradient on an Agilent 1200 series system: Step 1: 5–20% solvent D in in 6.25 min; Step 2: 20–90% solvent D in 52.5 min.

Product-containing fractions were pooled and further purified via anion exchange chromatography with a Dionex DNAPac PA100 column (250 mm × 4.0 mm) using a ternary solvent system containing acetonitrile (solvent E), 25 mM Tris*HCl pH 8 (buffer F) and 25 mM Tris*HCl 500 mM $NaClO_4$ (buffer G) and a linear gradient from 0% buffer G to 70% buffer G in 18.5 min while the content of solvent E was kept constant at 30%.

For all oligonucleotides, all product-containing fractions were concentrated, analysed by LC-MS (Bruker micrOTOF-Q), the purest fractions were pooled and desalted via NAP25 columns (GE Healthcare), lyophilized and dissolved in PBS buffer (80 ng μl$^{-1}$).

The sequences of the oligonucleotides used for in vivo studies were as follows: cholesterol-modified non-caged antimiR-92a: 5′-$C_sA_s$GGCCGGGACAAGUG-$CA_sA_sU_sA_s$-Chol-3′ (expected mass: 7939.7 Da, obtained mass: 7940.8 Da),

cholesterol-modified caged antimiR-92a: 5′-$C_sA_s$GG$C$CGGG$A$CAA$G$UG$C$$A_sA_sU_sA_s$-Chol-3′ (expected mass: 8834.0 Da, obtained mass: 8833.2 Da),

cholesterol-modified caged antimiR-Ctrl: 5′-$A_sA_s$GGC$A$AGCUGACCCUG$A$-$A_sG_sU_sU_s$-Chol-3′ (expected mass: 8518.8 Da, obtained mass: 8520.2 Da),

cholesterol-modified and Cy3-labelled caged antimiR-92a: 5′-Cy3-$C_sA_s$GG$C$-CGGG$A$CAAGUGC$A_sU_sA_s$-Chol-3′ (expected mass: 9340.2 Da, obtained mass: 9341.8 Da).

The subscript 's' represents phosphorothioate linkages, bold script represents caged nucleotides, 'Chol' represents the 3′-Cholesteryl-TEG modifier and 'Cy3' represents the cyanine 3 label.

**Animal experiments.** All animal experiments were performed in accordance with the animal ethics guidelines and were approved by the local authorities (Regierungspräsidium Darmstadt, Hessen, Germany). Skin from 12-week-old male C57/BL6N mice (Charles River, Germany) was shaved and LEDs with different wavelengths were tested. Light intensity behind the skin barrier was detected with a laser power meter 1918 and an 818P detector (Newport Spectraphysics). For testing light-inducible antimiR efficiency, shaved skin was disinfected and isolated for a 48 h tissue culture (1 cm$^2$ skin explants) and antimiRs (1 μg in 50 μl PBS) were injected intradermally. After irradiation for 10 min ($\lambda = 385$ nm, 300 mA, mounted UVLED-385-310 SMD with LED Driver DC2100 (Thorlabs)), skin explants were floated with cell culture medium (FAD medium: DMEM/Ham's F12 (3.5:1.1), low calcium (Biochrom) supplemented with 10% FCS (Invitrogen), $10^{-10}$ M cholera toxin (Sigma), 0.5 μg ml$^{-1}$ hydrocortisone (Sigma), 0.18 mM adenine (Sigma), 5 μg ml$^{-1}$ insulin (Invitrogen) and 10 ng ml$^{-1}$ EGF (Invitrogen)) for 48 h (5% $CO_2$, 37 °C).

For in vivo treatment, antimiRs were injected intradermally. Ten minutes after injection, the area was irradiated for 10 min ($\lambda = 385$ nm, 300 mA), while the remaining part of the body was protected with aluminum foil.

Thirteen-week-old male BKS(D)-Leprdb/JOrlRj (db/db) mice (Janvier Labs, France) were used for wounding experiments. Mice were anaesthetized by i.p. injection of Ketanest/Rompun (Ketavet, 100 mg kg$^{-1}$ bodyweight, Pfizer; Rompun 20 mg kg$^{-1}$ bodyweight, Bayer), and the back was shaved and disinfected. Two 6 mm full-thickness excisional wounds were created by using standard biopsy puncher (Stiefel, Germany). AntimiRs were applied topically into the open wound (2 μg in 25 μl). After 10 min, wounds were irradiated for 10 min ($\lambda = 385$ nm, 300 mA). Additionally, on days 4 and 7 postwounding, mice were treated with light-inducible antimiRs by intradermal injection into the wound margin (2 μg in 25 μl) as described above. The wound healing kinetics was documented photographically.

**Measurement for toxicity of light-inducible antimiRs.** Blood of light-inducible antimiR-treated mice (13-week-old male BKS(D)-Leprdb/JOrlRj (db/db) mice) was collected and plasma was isolated via centrifugation (15 min, 1,500g). As biomarkers for liver injury, plasma levels for transaminases aspartate aminotransferase and alanine aminotransferase as well as for alkaline phosphatase were quantified. Creatinine and urea plasma levels were determined to proof kidney functionality. Measurements occurred in accordance with standard procedures in the central laboratory of the university hospital Frankfurt am Main, Germany. Commercial cobas tests were used in accordance with the manufacturer's protocols (Roche Diagnostics).

**Preparation of human wound tissue samples.** Human tissue samples were obtained from patients presenting with healing (day 14–30 days postsurgery, $n = 6$) and non-healing (venous leg ulcers, $n = 6$) skin wounds. The tissue was fixed in 4% paraformaldehyde and paraffin embedded. RNA was isolated using the miRNeasy FFPE Kit (Qiagen) according to the manufacturer's instructions from 20 μm paraffin-embedded tissue sections. The use of human material was approved from the local ethics committee and informed consent of patients was received (University Hospital Cologne).

**Quantitative real-time RT-PCR.** Total RNA was isolated using the miRNeasy Mini Kit (Qiagen) according to the manufacturer's protocol. For miR-92a detection, 10 ng RNA was reverse transcribed and quantified by TaqMan real-time PCR using the TaqMan microRNA Assay (Assay name: hsa-miR-92, assay ID: 000430, Applied Biosystems). RNU6 was used as the endogenous control (Assay name: U6 snRNA, assay ID: 001973). A StepOnePlus device was used for detection (Applied Biosystems).

**Immunohistochemistry.** Immunohistochemical stainings were performed on 4 μm paraffin tissue sections or on frozen tissue sections (10 μm) in case of preceded Cy3-labelled antimiR-92a treatment. Tissue was blocked with 1% BSA/2% normal goat serum (Dako) in PBS and then incubated with the primary antibody overnight at 4 °C. Monoclonal primary antibodies used were: rat α mouse F4/80 (1:100, Abcam, no. ab6640), rat α mouse CD31 (1:30, Dianova, no. DIA-310), rabbit α mouse Itga5 (1:100, Abcam, no. ab150361), rabbit α mouse Ki67 (1:100, Abcam, no. ab15580), rat α mouse CD45 (1:100, Abcam, no. ab25386), and rabbit α mouse Sirt1 (1:100, Millipore, no. MABE426). Bound primary antibody was detected by incubation with Alexa Fluor 488-, Alexa Fluor 555- or Alexa Fluor 647-conjugated (Invitrogen) secondary antibodies (1:500, 1 h, room temperature), followed by counterstaining with Dapi. For αSMA staining, a directly Cy3-labelled antibody was used (1:400, Sigma, no. C6198).

**Morphometric analysis.** The macroscopic wound area was quantified based on photographs taken at various time points post injury and was calculated as the percentage of the initial wound area with a diameter of 6 mm. For scaling, a ruler was placed next to the wound. The extent of granulation tissue formation, as a measure for wound quality and stability, was determined on 4 μm haematoxylin and eosin-stained paraffin tissue sections from the central portion of the wound using a Zeiss microscope equipped with a bright field filter (Axio Observer.Z1, Zeiss). Additionally, the distance between the ends of panniculus carnosus was determined as a measure of wound contraction and the distance between ends of epithelial tips was measured for the wound closure rate. For the quantitative

analysis of antibody-stained tissue sections, images were generated with the laser scanning confocal microscope LSM 780 (Zeiss) at a magnification of 20 × or 25 × (oil). The percentage of the positive stained CD31, Itga5 and αSMA area was calculated by using the ImageJ software and normalized to the total wound area.

**Quantitative analysis of Sirt1 and Ki67 staining.** By using the ImageJ Software, the Sirt1 and Ki67-positive-stained area was measured and normalized to all Dapi-positive-stained nuclei within the wound area, including the hyperproliferative epithelium. CD45-positive cells were counted and normalized to the total wound area.

**Statistical analysis.** Significance of difference was analysed using analysis of variance one-way test analysis with Newman–Keuls Multiple Comparison Test. All data are presented as mean ± s.e.m.; a $P$ value of $\leq 0.05$ was considered significant.

**Data availability.** The authors declare that all data supporting the findings of this study are available within the paper and its Supplementary Information files.

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

## Acknowledgements

We thank Nicole Konecny, Marion Muhly-Reinholz, Felix Vetter, Yosif Manavski and Ariane Fischer for excellent technical support and Reinier A. Boon for helpful discussion. This work was supported by the German Research Society (SFB902 to S.D. and A.H., SFB829 and FOR2240 to S.A.E.), by the Leducq Network 'MIRVAD' to S.D. and by a start-up grant from the LOEWE center for cell and gene therapy, Frankfurt, Germany to T.L. and F.S.

## Author contributions

T.L. designed and performed experiments and analysed data. F.S and P.M. designed and synthesized light-inducible antimiRs and analysed data. S.A.E. provided human tissue samples. T.L., F.S., S.D. and A.H. designed experiments and wrote the manuscript.

## Additional information

**Competing interests:** S.D. holds a patent on miR-92a. The remaining authors declare no competing financial interests.

**Publisher's note**: 

