## [Peer review file · Nature Communications]

Reviewers' comments:

Reviewer #1 (Remarks to the Author):

A. Summary of the key results:

The authors previously developed light-activatable antimiRs, which can locally inhibit miRNA activity in vivo after light irradiation. In this study, they used this technique targeting miR-92a, which improved wound healing in a diabetic mouse model.

The technique of caged antimiR is very interesting and harbors great therapeutic potential, since this will avoid unwanted adverse effects in untargeted tissues during systemic delivery. This will be beneficial especially for the diseases in the well-perfused inner organs, e.g. acute myocardial infarction. However, when treating chronic wounds, the free antimiR-92 shows equally good therapeutic effects without obvious toxic side effects compared to the caged antimiR-92. Therefore, I think in topical treatment of skin wounds this technique does not show obvious advantages compared to the ordinary antimiR therapies.

B. Originality and interest: if not novel, please give references:

This study is original and it will be interesting to the researchers working in the fields of miRNA-based therapies, wound care etc.

C. Data & methodology: validity of approach, quality of data, quality of presentation:

The authors examined the inhibition efficiency of the caged antimiR-92 in the skin, liver and kidney of the wild-type mice (Fig 1B-D), which should also be done in the antimiR treated db/db wound models, since the different factors, e.g. skin injury, diabetic background, different doses and application interval of antimiR, may affect the efficiency and specificity of the caged antimiR-92. I think compared with the free antimiR, the major benefit of caged antimiR is that it may cause less unwanted side effects in the other organs. However, in the case of treating diabetic wounds, we do not know whether this is true.

The authors concluded that miR-92 a was up-regulated in chronic wounds, based on the data in Fig 2A. This is not appropriate, since only 3 human chronic wounds were included, and they were compared with the mice samples.

In line 330, the authors stated that 'Measurements occurred in accordance to standard procedures in the central laboratory...'. The methods to measure AST, ALT, AP etc. should be specified.

In line 505-507, the figure legends for Fig 2D and 2E were put in wrong order.

The pictures of immunostaining should be presented in the Supplementary figure 4, together the quantification results.

D. Appropriate use of statistics and treatment of uncertainties:

Yes

E. Conclusions: robustness, validity, reliability:

Please see my comments as above.

F. Suggested improvements: experiments, data for possible revision:

It would be interesting to test how long the inhibitory effect of caged antimiR last after irradiation? Is it possible to use doses lower than 2 μ g or inject less often (currently the antimiRs are applied on day 0, 4 and 7)?

How is caged antimiR distributed in the skin after intradermal injection? What cell types in the skin receive these antimiRs? Is the neighbouring skin affected?

G. References: appropriate credit to previous work:

Yes

H. Clarity and context: lucidity of abstract/summary, appropriateness of abstract, introduction and conclusions:

Yes

Reviewer #2 (Remarks to the Author):

Dimmeler, Heckel and coworkers describe the application of a light-activated oligo (antimiR) for the regulation of a specific miRNA, miR-92a, with previously established roles in controlling angiogenesis. To provide further analysis of miR-92a function in a rodent model of wound-healing, and to develop a potential antisense treatment against this miRNA target, the authors develop an in-vivo-stable form of a previously published antimiR vs. miR-92a, and investigate its light-dependent function in assisting wound healing.

There are several unique features of this work that warrant its publication in Nature Comm.

- 1) Validation of miR-92a as a therapeutic target for wound healing in a rodent model. The link to diabetic wound healing is made. This builds on refs 7, 12, 13 showing biological roles of this miRNA in angiogenesis.
- 2) First demonstration of a light-activated oligo as a potential therapeutic agent.
- 3) First demonstration of a light-activated oligo as a tool for assessing in vivo miRNA function.

4) First investigation of toxicity of photoactivated (caged) oligo. No toxicity/difference in gene expression was caused by photoactivation by the criteria established in this paper.

Other features of this work that require some examination:

1) Lack of clear benefit conferred from caging. Controls with uncaged anti-miR-92a show considerably more off-organ effects (in liver and kidney), which is a clear perturbation of the biological model system, even if toxicity of more systemic delivery is not in evidence. But is this a fair comparison? Structure of control compound anti-miR-92a is not provided in SI. Presumably this does not contain cholesterol, 2'-OMethylation, and phosphorothioation. Would similar 'targeting' be observed if compounds were identical, minus the nitrobenzyl moieties and photoactivation?

Authors should provide structure of this control compound in the SI.

Authors should consider performing control with identical (uncaged) compound, minus nitrobenzyls and photoactivation, to assess potential for targeting to site of injury when using i.d. injection.

Despite criticisms above (and actions recommended to correct some of them), I believe that this does not detract significantly from the work as the caged compound does allow a better test of the role of miRNA-92a in the specific tissue region undergoing wound healing. Very impressive knockdown of this miRNA is observed with both the control compound and the photoactivated compound. Observation of knockdown in liver and kidney with uncaged compound is a good demonstration for the importance of caging.

2) In a similar vein, this paper does not shed light on what the ultimate therapeutic agent for knocking down miR-92a should be--use of caged oligo may be unnecessary, particularly if the compound is delivered at the site of injury, intradermally.

However, this is arguably beyond the scope of this present work and could easily be the subject of a future manuscript, in a more medically oriented journal.

Nonetheless, the authors should provide their current thoughts on the most promising therapeutic strategy vs. this miRNA target in the conclusion. Is caged anti-miR just a tool, or also a practical and perhaps necessary therapeutic strategy in this case?

3) Relative lack of new chemistry. Caged anti-miR was developed in ref 8 by most of the same authors, and tested for ability to photomodulate angiogenesis in live cells.

In fact, the authors made several changes to the title compound to perform these in vivo studies.

*Ideally, the authors could show which, if any, of these modifications is important for

achieving in vivo activity.*

The authors should provide more characterization of their new caged compound, confirming identity and purity.

Finally, the authors did an excellent job in providing a very clear manuscript with text and figures that illustrate their key points and provide statistical significance for each. Conclusions are well supported. References are adequate.

Reviewer #3 (Remarks to the Author):

This manuscript describes the in vivo application of a photo-activateable antiMiR. The basic premise is that such an inducible system should reduce toxicity by only allowing function of the anti-MiR upon photo activation. The authors show efficacy of this approach in a wound healing model in which the antiMiR targets miRNA 92a. The miRNA is downregulated resulting in release of the miRNA target inhibition and expression of wound healing proteins. Overall, this is an innovative and carefully conducted study. A few points listed below should be addressed.

1. A figure showing the miR92a target sequence location and its complementarity to the miRNA should be included.
2. Does the antiMiR target only the mature miRNA?
3. Does the anti-miR get ingested by skin macrophages and travel via the blood stream to other body locations?
4. A figure showing the locations and structures of the photo activateable nucleobases in the anti-MiR would be a nice addition.
5. The authors should provide physical data showing release of the photo activateable reagents following photo activation.

ov

Point by point response

First of all, we would like to thank the Reviewers for their very helpful comments and overall their efforts, which helped us to substantially improve the manuscript.

Response to Reviewer #1

A. Summary of the key results: The authors previously developed light-activatable antimiRs, which can locally inhibit miRNA activity in vivo after light irradiation. In this study, they used this technique targeting miR-92a, which improved wound healing in a diabetic mouse model. The technique of caged antimiR is very interesting and harbors great therapeutic potential, since this will avoid unwanted adverse effects in untargeted tissues during systemic delivery. This will be beneficial especially for the diseases in the well-perfused inner organs, e.g. acute myocardial infarction. However, when treating chronic wounds, the free antimiR-92 shows equally good therapeutic effects without obvious toxic side effects compared to the caged antimiR-92. Therefore, I think in topical treatment of skin wounds this technique does not show obvious advantages compared to the ordinary antimiR therapies.

AntimiR-92a as a therapeutic agent to improve skin repair was chosen as proof-of-principle experiment, although it is known that systemic inhibition of miR-92a has less risks and side effects as in comparison to pro-regenerative antimiRs, which target tumor suppressor miRNAs such as miR-34a or miR-15 family members or cytotoxic antimiRs, which might be usable to treat skin cancer. Despite the limited risks of systemic antimiR-92a, our data support the concept that light-induced activation of antimiRs is feasible in vivo and can be therapeutically used to reduce systemic exposure of antimiRs. We hope that the reviewer agrees that establishing a second therapeutic strategy and model would extend the scope of the current manuscript. We revised the discussion section to highlight the potential of this novel approach in the revised manuscript.

B. Originality and interest: if not novel, please give references:

This study is original and it will be interesting to the researchers working in the fields of miRNA-based therapies, wound care etc.

C. Data & methodology: validity of approach, quality of data, quality of presentation:

The authors examined the inhibition efficiency of the caged antimiR-92 in the skin, liver and kidney of the wild-type mice (Fig 1B-D), which should also be done in the antimiR treated db/db wound models, since the different factors, e.g. skin injury, diabetic background, different doses and application interval of antimiR, may affect the efficiency and specificity of the caged antimiR-92. I think compared with the free antimiR, the major benefit of caged antimiR is that it may cause less unwanted side effects in the other organs. However, in the case of treating diabetic wounds, we do not know whether this is true.

To address the concern of the reviewer, we measured miR-92a expression in wounded db/db mice at day 6 and 11 post injury. As shown in Figure 1 for the reviewer, the inhibition of miR-92a expression in kidney and liver was more pronounced when constitutively active antimiRs were used compared to local light activated caged antimiRs. The effects observed in wounded db/db mice were thus comparable to healthy wild type mice (see comparison of the data provided in Figure 1 for the reviewer).

The authors concluded that miR-92a was up-regulated in chronic wounds, based on the data in Fig 2A. This is not appropriate, since only 3 human chronic wounds were included, and they were compared with the mice samples.

We included new qPCR measurements of acute human wound tissue and compared murine samples and human samples separately. For both species, we found a profound upregulation for miR-92a expression in chronic wounds in comparison to acute wound tissue. In addition we increased the sample size to n=6 for the human samples. These data are included in revised Figure 2A of the manuscript.

In line 330, the authors stated that 'Measurements occurred in accordance to standard procedures in the central laboratory...'. The methods to measure AST, ALT, AP etc. should be specified.

We used commercial kits from Roche Diagnostics. We add more details with respect to the methods used to determine liver and kidney toxic parameters in the material and method section (see page 12, paragraph 4 of the revised manuscript).

In line 505-507, the figure legends for Fig 2D and 2E were put in wrong order.

We apologize for the mistake and changed the order accordingly.

The pictures of immunostaining should be presented in the Supplementary figure 4, together the quantification results.

We provide representative images of the stainings in the revised Supp. Figure 7 for all groups. The new Supp. Figure 7B shows the α SMA staining and Supp. Figure 7D the CD45 staining.

D. Appropriate use of statistics and treatment of uncertainties: Yes

E. Conclusions: robustness, validity, reliability: Please see my comments as above.

F. Suggested improvements: experiments, data for possible revision: It would be interesting to test how long the inhibitory effect of caged anti-miR last after irradiation?

Based on a missing permission to keep the treated animals longer than 2 weeks, we have no data how long the inhibitory effect of caged anti-miRs lasts after irradiation of the skin or in wound tissue. Nevertheless based on our own experiences from former projects and from published data (Krutzfeld et al. Nature 2005)¹, we know that the effect of such anti-miRs is long lasting, for up to several weeks. For example, the anti-miR-92a used as positive control in this study efficiently suppressed miR-92a for 6 weeks after a single i.v. injection in heart and muscle tissue (see Figure 2 for the reviewer).

Is it possible to use doses lower than 2ug or inject less often (currently the anti-miRs are applied on day 0, 4 and 7)?

While establishing the experimental setup for our wound healing study, we initially treated the wounds just once on the day of wounding with 2 μ g/wound with caged anti-miRs. The data set shown in reviewer figure 3 shows already a trend of improved healing with only a single dose. Nevertheless, we decided to increase the dosage for the following experiments in order to achieve stronger effects.

How is caged anti-miR distributed in the skin after intradermal injection? What cell types in the skin receive these anti-miRs? Is the neighbouring skin affected?

To address this question, we performed a new set of in vivo experiments using Cy3-labelled caged anti-miR-92a to detect the fate of the injected anti-miRs. Cy3-labelled anti-miRs were intradermally injected using the same procedure as for the unlabeled anti-miRs (Figure 2) and cellular up-take was

visualized at 6 days post injury. We observed that antimiRs were taken up both in the hyperproliferative epithelium and the dermis. To identify immune cells, we counter-stained the sections with F4/80 and showed that macrophages are positive for Cy3-labelled antimiRs. These data are included in the new Supp. Figure 5 and in the text of the revised manuscript (page 6, paragraph 1).

G. References: appropriate credit to previous work: Yes

H. Clarity and context: lucidity of abstract/summary, appropriateness of abstract, introduction and conclusions: Yes

Response to Reviewer #2:

Dimmeler, Heckel and coworkers describe the application of a light-activated oligo (antimiR) for the regulation of a specific miRNA, miR-92a, with previously established roles in controlling angiogenesis. To provide further analysis of miR-92a function in a rodent model of wound-healing, and to develop a potential antisense treatment against this miRNA target, the authors develop an in-vivo-stable form of a previously published antimiR vs. miR-92a, and investigate its light-dependent function in assisting wound healing.

There are several unique features of this work that warrant its publication in Nature Comm.

- 1) Validation of miR-92a as a therapeutic target for wound healing in a rodent model. The link to diabetic wound healing is made. This builds on refs 7, 12, 13 showing biological roles of this miRNA in angiogenesis.
- 2) First demonstration of a light-activated oligo as a potential therapeutic agent.
- 3) First demonstration of a light-activated oligo as a tool for assessing in vivo miRNA function.
- 4) First investigation of toxicity of photoactivated (caged) oligo. No toxicity/difference in gene expression was caused by photoactivation by the criteria established in this paper.

We thank the reviewer to his/her comments highlighting the major novel findings of our study.

Other features of this work that require some examination:

- 1) Lack of clear benefit conferred from caging. Controls with uncaged antimiR-92a show considerably more off-organ effects (in liver and kidney), which is a clear perturbation of the biological model system, even if toxicity of more systemic delivery is not in evidence. But is this a fair comparison?

To address this question, we compared the inhibition of miR-92a in the main RNA-metabolizing organs kidney and liver after either injecting the same concentrations of systemically active antimiR-92a (as positive control), caged antimiR-92a without light exposure, and caged antimiR-92a that was activated by light compared to antimiR-control treated mice (see Figure 1 for the reviewer). In wild type mice, the constitutively active antimiR-92a significantly reduced miR-92a expression in the liver in comparison to the light activated antimiR-92a. In the kidney, the constitutively active antimiR-92a also showed a more efficient reduction in comparison to the light activated antimiR-92a. Similar results were obtained when db/db mice were used (right panel). The reason that light-activated antimiR-92a also have some miR-92a inhibitory effects in kidneys might relate to the fact that we cannot exclude light penetration in the tissue under this conditions (kidneys are directly beneath the injured wound area).

Structure of control compound antimiR-92a is not provided in SI. Presumably this does not contain cholesterol, 2'-OMethylation, and phosphorothioation. Would similar 'targeting' be observed if compounds were identical, minus the nitrobenzyl moieties and photoactivation?

Authors should provide structure of this control compound in the SI.

Authors should consider performing control with identical (uncaged) compound, minus nitrobenzyls and photoactivation, to assess potential for targeting to site of injury when using i.d. injection.

We apologize for the poor description of the compounds used. The structure of the control antimiR-92a, that we termed positive control, is identical to the caged antimiR-92a, which only has additional photolabile protection groups (see new Supplementary Figure 2). All antimiRs used for the therapeutic studies shown in Figure 2 and 3 contain cholesterol, 2'-OMe and the phosphorothioate backbone at the same positions. To illustrate this to the reader, we provided a revised Supp. Fig. 2 showing the design of all antimiRs used in the present study.

Despite criticisms above (and actions recommended to correct some of them), I believe that this does not detract significantly from the work as the caged compound does allow a better test of the role of miRNA-92a in the specific tissue region undergoing wound healing. Very impressive knockdown of this miRNA is observed with both the control compound and the photoactivated compound. Observation of knockdown in liver and kidney with uncaged compound is a good demonstration for the importance of caging.

2) In a similar vein, this paper does not shed light on what the ultimate therapeutic agent for knocking down miR-92a should be--use of caged oligo may be unnecessary, particularly if the compound is delivered at the site of injury, intradermally. However, this is arguably beyond the scope of this present work and could easily be the subject of a future manuscript, in a more medically oriented journal.

Nonetheless, the authors should provide their current thoughts on the most promising therapeutic strategy vs. this miRNA target in the conclusion. Is caged anti-miR just a tool, or also a practical and perhaps necessary therapeutic strategy in this case?

We agree with this comment. Given the low potential of cytotoxicity of systemic miR-92a treatment, our study can be rather seen as a “proof-of-concept” experiment pathing the way to testing cytotoxic or tumorigenic anti-miRs for therapeutic approaches (see also comment A to reviewer 1). As suggested, we comment on this issue in the discussion part of the revised manuscript.

3) Relative lack of new chemistry. Caged anti-miR was developed in ref 8 by most of the same authors, and tested for ability to photomodulate angiogenesis in live cells.

In fact, the authors made several changes to the title compound to perform these in vivo studies. *Ideally, the authors could show which, if any, of these modifications is important for achieving in vivo activity.*

For gaining sufficient in vivo efficiency, we additionally added cholesterol to the 3' end of the anti-miR, a strategy that was suggested by the pioneering work of Krutzfeld et al.¹ We showed that caging is also effective in such in vivo applicable anti-miRs. To make these modifications more clear to the reader, we added supplementary figure 2, which provides more details regarding the design of the anti-miRs.

The authors should provide more characterization of their new caged compound, confirming identity and purity.

We first determined the purity of the caged anti-miR-92a, non-caged anti-miR-92a and the light-activated anti-miR-92a by loading on polyacrylamide gel. As shown in the new Suppl. Figure 3A, all three substances showed one single band, with caged anti-miR-92a showing a higher molecular weight. To confirm the purity we additionally used analytical HPLC. As shown in new supplementary figure 3B, we detected only one major peak in all three preparations. Furthermore, caging lead to a shift in the peak, which was reversed by light induction.

Furthermore, we determined the identity of all oligonucleotides by ESI-LC/MS. All oligonucleotides showed one major peak with the expected size (Suppl. Figure 3C).

Finally, the authors did an excellent job in providing a very clear manuscript with text and figures that illustrate their key points and provide statistical significance for each. Conclusions are well supported. References are adequate.

Response to Reviewer #3:

This manuscript describes the in vivo application of a photo-activateable antiMiR. The basic premise is that such an inducible system should reduce toxicity by only allowing function of the anti-miR upon photo activation. The authors show efficacy of this approach in a wound healing model in which the antiMiR targets miRNA 92a. The miRNA is downregulated result in release of the miRNA target inhibition and expression of wound healing proteins. Overall, this is an innovative and carefully conducted study. A few points listed below should be addressed.

1. A figure showing the miR92a target sequence location and its complementarity to the miRNA should be included.

As suggested by the reviewer, we included miR-92a target sequence and its complementarity to the microRNA in Figure 4 for the reviewer. Furthermore we added the exact sequence of our antimiRs as well as for miR-92a in Supp. Figure 2.

2. Does the antiMiR target only the mature miRNA?

The antimiRs only reduced the mature miR-92a but did not affect the expression of the pri-miR (see figure 5 for the reviewer).

3. Does the anti-miR get ingested by skin macrophages and travel via the blood stream to other body locations?

In order to analyze the distribution of caged antimiRs, we injected Cy3-labelled caged antimiR-92a and isolated the wound tissue 6 days post injury (Suppl. Figure 5). Positive cells were detected in the hyperproliferative epithelium and within the dermal compartment. Moreover, double staining showed that Cy3-labelled antimiRs are taken up by F4/80-positive macrophages (Suppl. Figure 5). This indicates that macrophages indeed take up the antimiR. However, we cannot explore whether the loaded macrophages can transfer the antimiR to the blood stream or other organs.

4. A figure showing the locations and structures of the photo activateable nucleobases in the anti-MiR would be a nice addition.

As suggested by the reviewer, we provide the design of the antimiRs in Supplemental Figure 2 of the revised manuscript, showing the location and structure of the photo activatable nucleobases.

5. The authors should provide physical data showing release of the photo activateable reagents following photo activation.

To address this question, we performed analytical HPLC experiments showing that indeed light activation induces the release of the caging groups (see Suppl. Figure 3A-C).

Figure 1 for the reviewer: qPCR analysis of miR-92a expression in skin, liver, kidney or wound tissue. Left, tissue isolated from unwounded Bl/6 wildtype animals treated once with antimir-92a as indicated. Right, tissue isolated from wounded db/db mice, treated with antimir-92a as indicated at day 0, 4 and 7 post injury.

Figure 2 for the reviewer: qPCR analysis of miR-92a expression level in heart and muscle tissue 6 weeks after one single injection of antimir-92a as indicated.

Figure 3 for the reviewer: Left, qPCR analysis of miR-92a expression level in wounded db/db mice that received a single treatment of antimir-92a (2 µg/wound) as indicated. Right, morphometric quantification of wound healing parameters after a single treatment with antimir-92a.

hsa-miR-92 binding site in SIRT1 3'UTR (miRANDA database)

Figure 4 for the reviewer: complementary target sequence of ITGA5 (top panel) and Sirt1 (bottom panel).

Figure 5 for the reviewer: qPCR measurement of pri-miR expression in caged anti-miR-92a treated wounds without and with light activation (n = 3).

Reference

(1) Krützfeldt J, Rajewsky N, Braich R, Rajeev KG, Tuschl T, Manoharan M, Stoffel M. (2005) Silencing of microRNAs in vivo with 'antagomirs'. Nature 438(7068):685-9.

REVIEWERS' COMMENTS:

Reviewer #1 (Remarks to the Author):

The authors answered all the questions I raised and I do not have further comments.

Reviewer #2 (Remarks to the Author):

Authors have successfully addressed all previous critiques in this revised manuscript. This work is now suitable for publication in this format.

Reviewer #3 (Remarks to the Author):

The authors have satisfactorily addressed the reviewer concerns. This is an important contribution to therapeutic oligonucleotide technology